# Two novel mutations in *MSX1* causing oligodontia

Le Yang☯, Jia Liang☯, Haitang Yue, Zhuan Bian* 

The State Key Laboratory Breeding Base of Basic Science of Stomatology (Hubei-MOST) and Key Laboratory of Oral Biomedicine Ministry of Education, School and Hospital of Stomatology, Wuhan University, Wuhan, Hubei Province, China

☯ These authors contributed equally to this work.
* bianzhuan@whu.edu.cn

## Abstract

Tooth agenesis is one of the most common developmental anomalies in humans and can affect dental occlusion and speech pronunciation. Research has identified an association between mutations in *MSX1*, *PAX9*, *EDA*, *AXIN2*, *WNT10A*, *WNT10B* and *LRP6* and human tooth agenesis. Two unrelated individuals with non-syndromic tooth agenesis and their families were enrolled in this study. Using Sanger sequencing of the candidate genes, we identified two novel mutations: a missense mutation c.572 T>C and a frameshift mutation c.590_594 dup TGTCC, which were both detected in the homeodomain of MSX1. After identifying the mutations, structural modeling and bioinformatics analysis were used to predict the resulting conformational changes in the MSX1 homeodomain. Combined with 3D-structural analysis of other *MSX1* mutations, we propose that there is a correlation between the observed phenotypes and alterations in hydrogen bond formation, thereby potentially affecting protein binding.

## Introduction

Tooth agenesis is a common developmental anomaly of the human dentition. In the general population, the incidence of tooth agenesis of permanent teeth ranges from 2.2% to 10.1% [1]. Several terms are used to describe tooth agenesis: hypodontia is referred as an absence of less than six teeth excluding the third molars; oligodontia is an absence of six or more teeth excluding the third molars; and anodontia is the complete absence of teeth [2]. Tooth agenesis is classified as syndromic when it is accompanied by other inherited abnormalities including nail dysplasia [3], sparse hair, lack of sweat glands or a cleft lip and palate [4–8], and classified as non-syndromic when the absence of teeth is an isolated characteristic. Tooth agenesis occurs either in a sporadic or in a hereditary manner, in autosomal dominant [9], autosomal recessive [10] or X-linked patterns [11]. The etiology of tooth agenesis is complex and not yet completely elucidated [2]. Environmental factors potentially influencing the dental development include trauma, chemotherapeutic drugs, radiotherapy or thalidomide use during pregnancy [12, 13]. The majority of cases are caused by genetic mutations, and to date, mutations in the *MSX1*, *PAX9*, *AXIN2*, *WNT10A*, *EDA*, *EDAR*, *EDARADD*, *WNT10B* and *LRP6* genes have been associated with non-syndromic tooth agenesis (NSTA) cases [14–22].

**Data Availability Statement:** All relevant data are within the paper and its Supporting Information files.

**Funding:** This study was supported by grants from the General Program of National Natural Scientific Foundation of China (No. 81970923), National Key Research and Development Program of China (No.

2016YFC1000505), Key Research Program of Provincial Department of Science and Technology (No. 2017ACA181), and Natural Science Foundation of Hubei Province (No. 2017CFB212).

**Competing interests:** The authors have declared that no competing interests exist.

*MSX1* and *PAX9* are the first genes identified causing NSTA [17, 19]. Both genes are transcription factors that play crucial part during the bud to cap stages of odontogenic development. Msx1 and Pax9 can interact with each other and they also act synergistically to activate the Bmp4 which is critical for tooth development [23]. EDA, EDAR and EDARADD are candidate genes of both NSTA and STA. EDA/EDAR/EDARADD signaling has been shown to play an important role in NSTA [24]. Recently, more and more genes in Wnt signaling pathway have been found to be related to NSTA. The first gene of Wnt signaling pathway involved in NSTA was *AXIN2* [20]. This gene was also found to be associated with a variety of cancers [25]. Axin2 plays a critical role in regulating the stability of β-catenin in the Wnt signaling pathway [26]. The second gene was *WNT10A* which encodes a secreted signaling protein. *WNT10A* mutations have been reported to be linked to a majority of NSTA cases [16]. Wnt10a is a key mediator of Wnt signaling which is required for proper tooth development [27]. In 2015, mutations in the WNT co-receptor *LRP6* were identified in families with autosomal-dominant non-syndromic oligodontia [15]. In 2016, WNT10B mutations causing oligodontia had been reported in a Chinese population [14].

Among those genes, NSTA is mostly associated with mutations in *PAX9*, *MSX1*, *WNT10A*, *AXIN2 and EDA* [28–30]. Therefore, in this study, we searched for mutations in the above five genes in two unrelated individuals with non-syndromic oligodontia. The study objectives were: 1) to identify the mutations responsible for the tooth agenesis in two Chinese patients; 2) to use 3D-structure modeling to explain the reasons why these mutations can lead to tooth agenesis.

## Materials and methods

### Subjects

Two unrelated Chinese oligodontia patients were identified in the orthodontic department at the School of Stomatology, Wuhan University. Intraoral examination and panoramic radiographs were taken to verify the number and location of missing teeth of each patient. Pedigree construction was achieved using clinical examinations and verbal interviews with the available family members. Blood samples were collected from the probands, their available family members and 300 unrelated healthy controls. All procedures were approved by the National Natural Science Foundation of the Medical Ethics Committee of the Stomatological Hospital of Wuhan University, China (Ethics Approval Identifier: A 42, date of approval: March 4, 2019). All participants in this study gave their written informed consent to publish these case details.

### Identification of mutations

Genomic DNA was extracted from the peripheral blood samples of all consenting family members and controls according to standard salt extraction procedures [31]. Screening of pathogenic mutations was performed using polymerase chain reaction (PCR) amplification and sequencing the complete exons and exon–intron boundaries of these five genes: *MSX1* (ENSG00000163132), *PAX9* (ENSG00000198807), *EDA* (ENSG00000158813), *AXIN2* (ENSG00000168646) and *WNT10A* (ENSG00000135925).

### Multiple sequence alignments

Multi-species amino acid sequence alignment of the MSX1 protein sequence (NP_002439.2) was performed using Clustal X (https://www.ebi.ac.uk/Tools/msa/). MSX1 sequences from dolphin to human were obtained from ENSEMBL.

## 3D modeling of MSX1 variants

On the basis of the crystal structure of the MSX1 homeodomain complex with DNA [32], the three-dimensional structure of the wild-type MSX1 homeodomain was derived using SWISS--MODEL (https://www.swissmodel.expasy.org) with PDB: 1ig7.1.C as a template. The 3D model was constructed using Swiss Pdb Viewer V 4.1 [33]. Visualization of the three-dimensional (3D) structures was performed with PyMOL (The PyMOL Molecular Graphics System, Version 0.99rc6, Schrödinger, LLC., Cambridge, MA, USA).

## Statistical analysis

Differences between the average number of missing teeth caused by missense mutations in the *MSX1* homeodomain with or without alterations in hydrogen bonding were analyzed using an unpaired, two-tailed *t*-test with GraphPad Prism 5 (GraphPad Software Inc., La Jolla, CA, USA). P-value < 0.05 was considered significant.

# Results

## Pedigree and phenotype analysis

Pedigree analysis of the two families revealed that the proband (II2) in family 1 was a sporadic patient and oligodontia in family 2 was exhibited in an autosomal dominant manner (Fig 1A). The proband (II2) in family 1 was a 16-year-old female with the congenital absence of ten permanent teeth excluding third molars. The second upper deciduous teeth were retained (Fig 1B and 1C), and both of her parents and her elder brother had normal dentition. The proband (III1) in family 2 was a 21-year-old male with more severe tooth agenesis, characterized by the absence of twenty permanent teeth excluding the third molars. The shape of 25 was conic, and most deciduous teeth were retained (Fig 1B and 1C). The mother of the proband (II2) was also diagnosed as tooth agenesis through oral examination. However, she could not recall or confirm extraction of some teeth. No orofacial cleft or other craniofacial abnormalities were noted in the affected members of the two families. Additionally, all reported subjects had normal primary dentition, nails, skin, and hair.

## Mutation analysis

Mutation analysis of candidate genes detected two novel heterozygous mutations in *MSX1* in two separate families. The proband of family 1 demonstrated a missense mutation at c.572T>C (Fig 2A). Her parents were wild-type at c.572, indicating a *de novo* mutation. A cross-species alignment of the MSX1 protein showed that p. Phe191 is highly conserved in the homeodomain (Fig 2B). This mutation replaced the hydrophobic phenylalanine with a hydrophilic serine at amino acid position 191. Screening of candidate genes in the proband of family 2 showed a frameshift insertion c.590_594 dup TGTCC in *MSX1* (Fig 2C). His mother also carried the mutation. This mutation is occurred in the homeodomain of MSX1, introducing 20 novel amino acids following the first 198 amino acids, while deleting the normal protein sequence from Ile199 through The303. Neither mutation was detected in the control group and had not been reported by 1000 Genomes, dbSNP, Human Gene Mutation Database (professional version) or PubMed. No pathogenic mutations were found in the *PAX9*, *AXIN2*, *EDA* and *WNT10A* genes of these two patients.

## Bioinformatics analyses

Secondary structural analysis showed that the homeodomain of the wild-type MSX1 protein consisting of 60 amino acids (No.172-231) is composed of the N-terminal arm (NT arm,

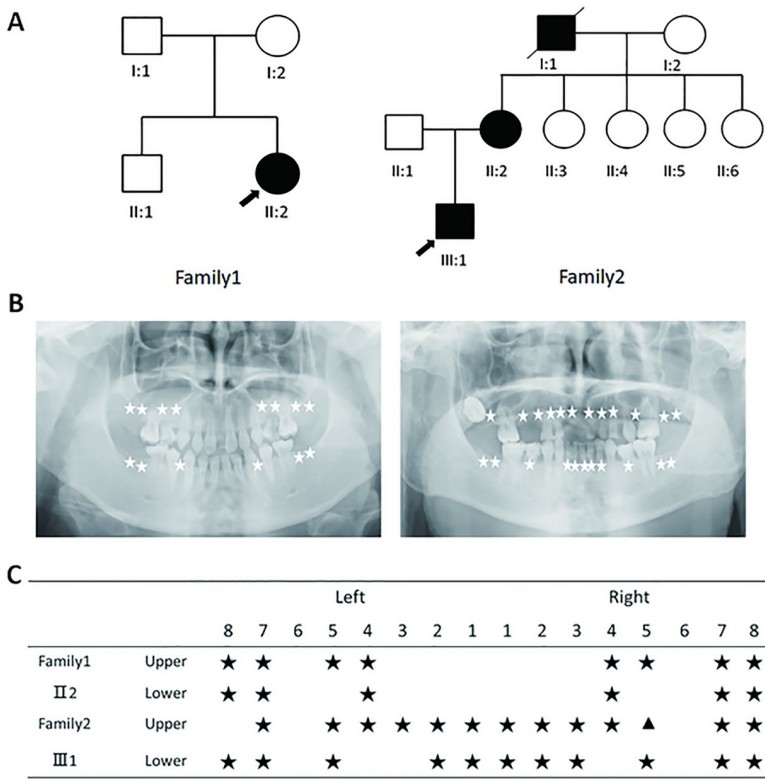

**Fig 1. Identification of two Chinese families with non-syndromic tooth agenesis.** (A) Pedigrees. Filled and unfilled symbols indicate affected and unaffected individuals, respectively. Squares and circles represent males and females, respectively. (B) Panoramic radiograph of the probands. The white stars indicate missing permanent teeth. Left: The proband of family I. Right: The proband of family II. (C) Tooth phenotypes of the probands with oligodontia. Stars indicate congenital missing tooth and triangle denotes cone-shaped teeth.

No.172-180) and three α-helices: helix I (No.181-193), helix II (No.199-209) and helix III (No.213-231). The mutation p. Phe191Ser was located at the end of the helix I (Fig 3A and 3B). In the wild-type MSX1 protein, the phenyl ring of the p. Phe191 was larger and overlapped with p. Gln195 (Fig 3A). In the p. Phe191Ser-mutated MSX1 protein, the phenyl ring disappeared and resulted in a large gap between p.191Ser and p. Gln195 (Fig 3B). Therefore, the p.Phe191Ser mutation may result in an abnormal structure of the MSX1 protein. In the wild type homeodomain, modelling predicted one hydrogen bond between Phe191 and Leu187 (Fig 3C), while in the mutated protein, there are two additional hydrogen bonds to Glu188 (Fig 3D).

The frameshift insertion (c.590_594 dup TGTCC, p.L197SfsX22) is located at the end of the loop domain between helix I and II, and generates a premature stop codon after an unrelated polypeptide sequence consisting of 22 amino acid residues. The shortened mutant-MSX1 protein is composed of two α-helices: helix I (No.181-193), and helix II (No.200-204), only the first helical region (helix I) was conserved and the other two helical regions (helixes II and III) are disrupted (Fig 3E).

## Discussion

*MSX1* is the first gene identified causing NSTA, which encodes a transcription factor [19]. It is widely expressed in many organs, particularly during the bud and cap stages of tooth development where epithelial–mesenchymal interactions occur in odontogenesis [34]. *MSX1* consists of two exons, the second of which includes the highly conserved homeodomain (HD). The

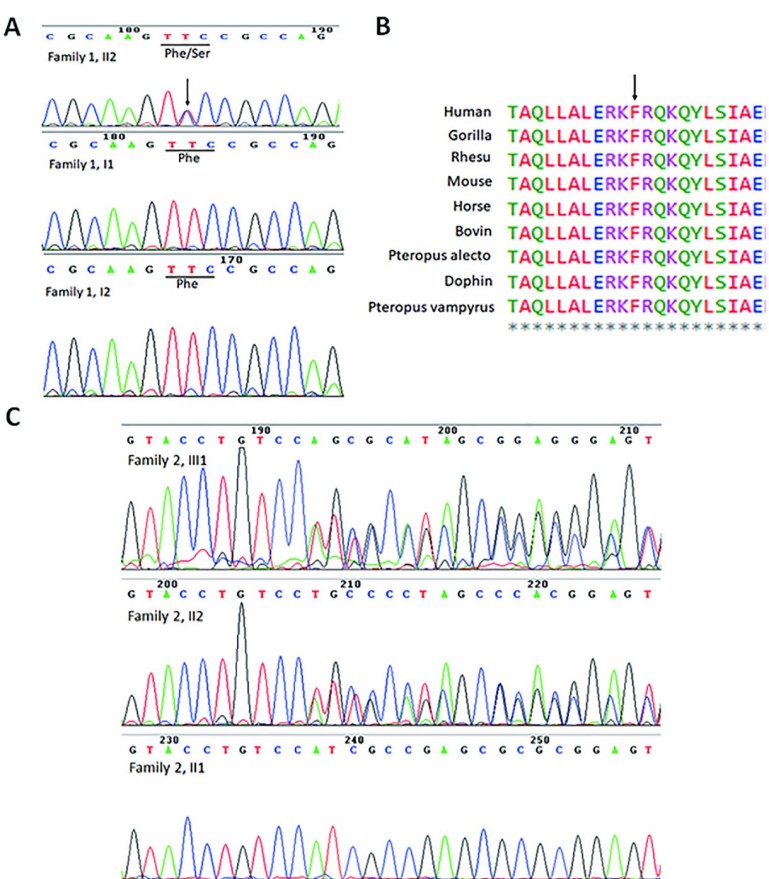

**Fig 2. *MSX1* mutations isolated from tooth agenesis patients.** (A) DNA sequence chromatograms presenting a *de novo* heterozygous missense mutation c.572T>G in *MSX1* identified in family I, compared with wild-type control. (B) Conservation analysis shows that the Phe residue at 191 in Msx1 is conserved across human, gorilla, rhesus, mouse, horse, bovin, pteropus alecto, dophin, pteropus vampyrus. (C) DNA sequence chromatograms showing a heterozygous c.590_594 dup TGTCC mutation identified in family II, compared with wild-type.

homeodomain is comprised of an extended N-terminal arm and three α-helices [35]. The residues in helices I and II are thought to play an important role in structural stability and binding activity while the residues in helix III combined with those in the N-terminal arm are important for DNA binding specificity[36]. It has been demonstrated that the homeodomain is important for the successful interaction of *MSX1* with other proteins such as TATA box binding protein (TBP) and distal-less homeobox (DLX) [37, 38]. However, the role of *MSX1* in human craniofacial and dental development has not been fully elucidated [39].

As we know, proteins fold into specific configurations mainly through a large number of non-covalent interactions, including hydrogen bonds, ionic bonds, van der Waals forces and hydrophobic interactions [40]. Hydrogen bonding plays a critical role in secondary structure formation and the integrity of three-dimensional structures [41]. In the homeodomain of MSX1, the stable secondary helical structure is maintained by hydrogen bonds between C = O and N-H groups of different amino acids. In this study, we show that the missense mutation Phe191Ser may lead to two additional hydrogen bonds with Glu188. Moreover, the mutated side chain becomes smaller, resulting in an increased distance between p.191Ser and p.Gln195. This alteration may force the neighboring residues to rearrange their positions, compromising the stability of the homeodomain and affecting the binding ability of MSX1 to DNA.

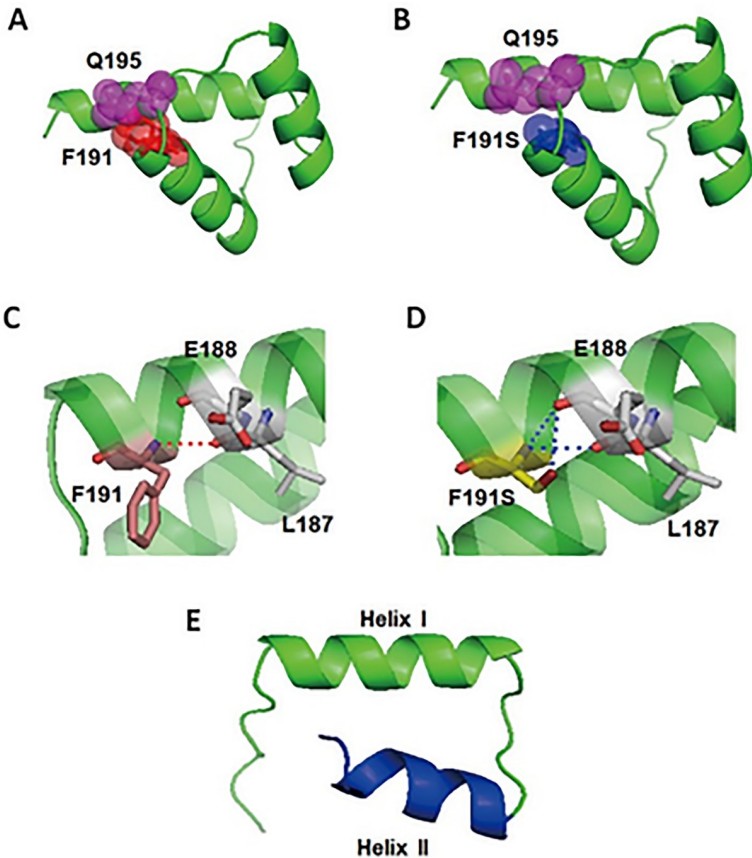

**Fig 3. Structural modeling of the wide-type and the mutated homeodomain of MSX1 protein.** (A, B) Structural modeling of the wide-type and the p. Phe191Ser mutated homeodomain of MSX1 protein show that the mutation is located at the end of the helix I. (A) In the wild-type MSX1 protein, the phenyl ring of the p. Phe191 is large and overlapped with p. Gln195. (B) In the p. Phe191Ser-mutated MSX1 protein, the phenyl ring disappears, a large gap between p.191Ser and p.Gln195 arise. (C) Phe191 forms a hydrogen bond with Leu187. (D) The Phe191Ser mutation creates another two new hydrogen bonds with Glu188. (E) Structural modeling of the c.590_594 dup TGTCC mutated homeodomain of MSX1protein by Pymol. The mutant structural shows that helix I is conserved, but helixes II and III are disrupted.

In order to elucidate the relationship between the hydrogen bond modifications induced by *MSX1* mutations and hypodontia phenotypes, we performed homology modeling to analyze the distribution of mutation sites in the homeodomain. We included 7 *MSX1* missense mutations from all missense/nonsense MSX1 mutations in HGMD (Human Gene Mutation Database, professional version 2019.3) [42], which are believed to cause non-syndromic tooth agenesis (Table 1)(S2 Table) (S2 Fig). Large deletions and insertions, small in-frame deletions, and nonsense mutations were not included because these mutations would damage the overall structure of MSX1 protein.

Our structural modeling predicts a correlation between observed phenotypes and the hydrogen bond alterations. *MSX1* mutations that altered hydrogen bonding tend to cause a more severe NSTA phenotype with more missing teeth (the average number of missing teeth were 3.8 vs. 8.8) (S3 Table) (Fig 4). Although hydrogen bonds are not the tractive force of protein or peptide folding, these bonds significantly contribute to the maintenance of the peptide folding [47]. Missense mutations that affect hydrogen bonds may have a significant effect on protein stability, DNA binding specificity, protein expression and interactions. Therefore, it is

**Table 1. MSX1 homeodomain missense variants in subphenotypes of increasing severity of non-syndromic tooth agenesis (NSTA) and alterations in hydrogen bonds.**

| MSX1 variants (HGVS; cDNA) | MSX1 variants (HGVS; predicted protein) | Missing teeth no. | Average no. of missing teeth | No. of patients | Phenotype | Alterations in hydrogen bonds | location | Reference |
|---|---|---|---|---|---|---|---|---|
| c.539C>T | p.(T180I) | 5 | 5 | 1 | Sporadic form of hypodontia | No | NT arm | [43] |
| c.689T>C | p.(L230P) | 2–4 | 3.6 | 5 | Autosomal dominant, non-syndromic hypodontia | No | Loop 3 | [44] |
| c.572 T>C | P.(F191S) | 10 | 10 | 1 | Sporadic form of oligodontia | Add | Helix I | This study |
| c.605G>C | p.(R202P) | 4–11 | 6.8 | 9 | Autosomal dominant tooth agenesis | Reduce | Helix II | [19] |
| c.632T>G | p.(L211R) | 8–18 | 11.7 | 3 | Autosomal dominant, non-syndromic oligodontia | Add | Loop 2 | [43] |
| c.673G>Ac | p.(A225T) | 11–19 | 15 | 2 | Autosomal recessive oligodontia with dental anomalies | Add+ Reduce | Helix III | [45] |
| c.680C>A | p.(A227E) | 5–13 | 8 | 4 | Autosomal dominant, non-syndromic, oligodontia | Add | Helix III | [46] |

likely that the mutations we report in this study resulted in more severe selective tooth agenesis. Further functional analysis of these mutations will help to reveal the molecular mechanism of their action.

In our study, a novel frameshift insertion of 5 bp (TGTCC) in the homeodomain of the *MSX1* gene (NM_002448) was also identified in a Chinese family with autosomal dominant non-syndromic tooth agenesis, affecting the amino acid sequence of the homeodomain at

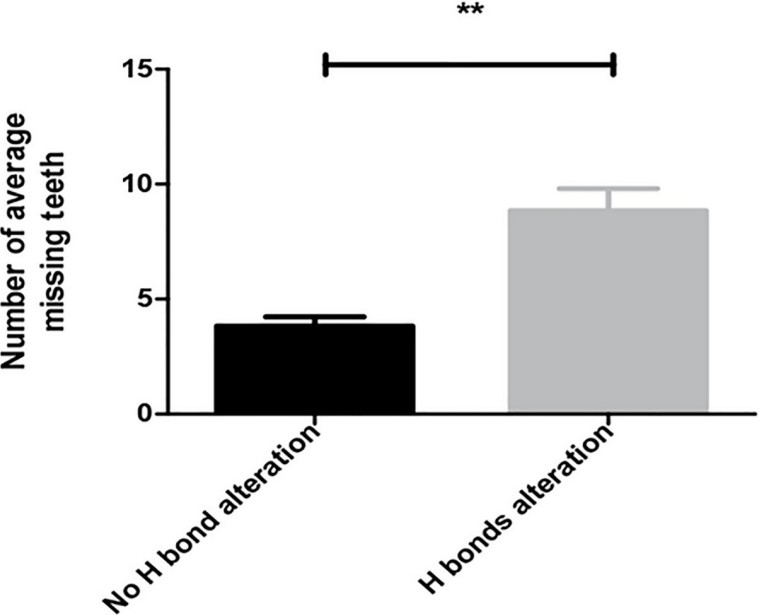

**Fig 4. Statistical analysis.** Comparison of the average missing teeth number caused by missense mutations in *MSX1* homeodomain with and without hydrogen bonds alteration obtained from different non-syndromic tooth agenesis patients (patients caused by missense mutations that had no hydrogen bond alteration, $n = 6$, vs. patients caused by missense mutations that had hydrogen bonds alteration, $n = 19$). Data are shown as mean ± SEM. An unpaired *t* test with two-tailed *p*-values was performed using GraphPad Prism 5 software. P value = 0.0086 obtained by analyzing data. **denotes a *P*-value <0.01.

p. 197Leu, changing the amino acid sequence from position 199. This alteration changes the spiral shape of helix II and eliminates the helix III, which is important for DNA-binding and protein-protein interactions including the TBP and DLX families [48]. Previous studies have shown that consensus residues that interact with DNA are located in helix III and the N-terminal arm [35]. We speculate that this novel frameshift mutation has a significant impact on the structure of the truncated protein, leading to severe consequences in protein interactions and DNA binding.

This frameshift mutation likely has considerable significant downstream effects as it demonstrates a substantial impact on primary protein structure. The MSX1 homeodomain contacts its target DNA regions in the major and minor groove at helix III and the N-terminal arm, respectively. The insertion variant found in the present study is predicted to be harmful through structural modeling and bioinformatics analyses, and is predicted to affect the DNA-binding functions of the homeodomain. Further studies are required to investigate the subcellular localization of this mutant MSX1 protein *in vitro*, to verify whether this is a functionally null variant.

## Conclusions

In our current study, we described two novel variants of the *MSX1* gene identified in two Chinese patients with isolated tooth agenesis: c.572T>C and c.590_594 dup TGTCC. These two novel mutations were proven to be pathogenic with respect to amino acid structure. These results expanded the mutational spectrum of the *MSX1* gene and the options to be considered for their precision treatment later. Further expression and functional studies are required to assess the effect of the identified mutations at the protein level.

## Supporting information

**S1 Fig. Comparison of wild type and c. 590_594 dup TGTCC protein sequences of the MSX1 homeodomain underlined.**
(TIF)

**S2 Fig. Predicted mutational impacts on the MSX1 homeodomain structure.** Thr180Ile and Leu230Pro all caused less than 6 missing teeth, neither mutation seems to cause any obvious changes in the structural interface, possibly explaining the milder effects. The other mutations all caused more than 6 missing teeth and had alterations in hydrogen bond formation. The alterations may lead to changes of the helical conformation, ultimately resulting in an alteration of protein folding and decreased stability.
(A) Locations of the mutational sites on the structural model of human MSX1 homeodomain structure (MSX1 homeodomain green, mutational sites red).
(B)- (G) Pair-wise comparisons between the wild-type (left) and mutant (right) residues for predicted changes in local contacts with other amino acids. The salmon stick models indicate wild-type residues, yellow indicate mutated residues, gray indicate neighboring residues, with the short bar in white, red and blue standing for the carbon, oxygen and nitrogen atoms, while the dashed lines in red and blue illustrated the hydrogen bonds. Here, we define a hydrogen bond geometrically as having a donor–acceptor distance ≤3.3 Å. The hydrogen bond distances are determined by Swiss-pdb Viewer.
(B) Leu230 and the Leu230Pro mutation both do not have hydrogen bond interactions with adjacent residues. So, the mutation does not seem to cause any obvious changes in the structural interface, possibly explaining the milder effects found for this mutation.
(C) Thr180 is predicted to have hydrogen-bonding interactions with Gln183 and Leu184. The

Thr180Ile mutation does not alter hydrogen-bonding with these two amino acids.

(D) Arg202, located in the α helix II, forms hydrogen bonds with adjacent residues Ser206 located in the same α helix II and Ser198 in the loop and Glu213 in the α helix III. The p. Arg202Pro mutation is predicted to abolish the hydrogen bonding with Ser198 and Glu213.

(E) Leu211 is predicted to have hydrogen-bonding interaction with Ser206 located in the same α helix II. The mutant protein has a longer side chain than the wild-type protein. The Leu211Arg introduce two new hydrogen bonds with Ser210 in the loop.

(F) Ala225 forms hydrogen bonds with adjacent residues Gln221, Asn222 and Arg229, The p. Ala225Thr mutation appears to eliminate the hydrogen bond with Asn222, but creates a new hydrogen bond with Gln221 in the same a-helix.

(G) Ala227 forms hydrogen bonds with adjacent residues Arg223 and Arg224, the p.Ala227-Glu mutation appears not to affect these hydrogen bonds but introduce two new hydrogen bonds with Arg223 and Leu230.

The figures are prepared using Pymol.

(TIF)

**S1 Table. Primers for candidate genes.**
(DOCX)

**S2 Table. Missense /nonsense MSX1 mutations from HGMD (2019.3).**
(DOCX)

**S3 Table. The (average) number of patients and missing teeth in *MSX1* mutations with and without hydrogen-bonding alteration.**
(DOCX)

## Acknowledgments

The authors of this study would like to thank the patients and their families for their important contribution to this study.

## Author Contributions

**Conceptualization:** Le Yang, Zhuan Bian.

**Data curation:** Le Yang, Zhuan Bian.

**Formal analysis:** Zhuan Bian.

**Funding acquisition:** Jia Liang.

**Investigation:** Jia Liang, Haitang Yue.

**Methodology:** Jia Liang, Haitang Yue.

**Project administration:** Jia Liang, Haitang Yue.

**Resources:** Jia Liang.

**Software:** Le Yang, Jia Liang.

**Supervision:** Le Yang, Zhuan Bian.

**Validation:** Jia Liang, Zhuan Bian.

**Visualization:** Le Yang, Jia Liang, Zhuan Bian.

**Writing – original draft:** Le Yang, Jia Liang.

**Writing – review & editing:** Jia Liang, Zhuan Bian.

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
