## [Decision Letter · Decision Letter 0]

14 Oct 2019

PONE-D-19-23967

Two Novel Mutations in MSX1 Causing Oligodontia

PLOS ONE

Dear Dr. Bian,

Thank you for submitting your manuscript to PLOS ONE. After careful consideration, we feel that it has merit but does not fully meet PLOS ONE’s publication criteria as it currently stands. Therefore, we invite you to submit a revised version of the manuscript that addresses the points raised during the review process by two reviewer's comments and suggestions.

This editor has additional suggestions:

In abstract: page 2, line 12: it seems the “with” should be replaced with “between”;In abstract: page 2, line 22: “ultimately” can be replaced with “thereby potentially”; In the Introduction part, 2^nd^ paragraph (dealing with the MSX1 gene) could be moved to Discussion part.  Instead, briefly introducing signaling and genetic contributions of each of nine genes (mentioned in the 1^st^ paragraph) to tooth agenesis.In the Introduction part, 3rd paragraph: please give reasons why the five genes (*MSX1*, et al.) are selected to screen the affected individuals.  In Results part, page 8, line 141: please add the range of homeodomain in amino acid residue numbers (from amino acid residual 172 to 234?).Based on current version of HGMD, the number of different missense mutations in the HOX domain should be at least 12, not 7.Page 10, line 186-196, figure 3C-3D of this part can be included as a supplementary figure.In Discussion part, page 11, line 205-206, replace “demonstrate” with “show” and “leads” with “may lead”.Page 11, line 211-212, the statement in 2016 may not be true any longer.  Please check current professional version of HGMD.Page 11, line 215-220, again, the number missense mutations in the HOX domain is more than 7. Please check current professional version of HGMD, and try to include all missense alleles.Figure 4 can be used as supplementary figure.Suggestion: Please consider to have a phenotypic comparison of all missense-related in non-HOX domain to the Table 3 results. If it turns out to be significant, Table 2 as a summary can be added to the main-text.

We would appreciate receiving your revised manuscript within 8 weeks. To enhance the reproducibility of your results, we recommend that if applicable you deposit your laboratory protocols in protocols.io, where a protocol can be assigned its own identifier (DOI) such that it can be cited independently in the future. For instructions see: http://journals.plos.org/plosone/s/submission-guidelines#loc-laboratory-protocols

We look forward to receiving your revised manuscript.

Kind regards,

Tao Cai, M.D., Ph.D.

Academic Editor

PLOS ONE

Journal Requirements:

2. Please provide additional details regarding participant consent. In the ethics statement in the Methods and online submission information, please ensure that you have specified whether your study included minors, and if so, state whether you obtained consent from parents or guardians. If the need for consent was waived by the ethics committee, please include this information.

Reviewers' comments:

Reviewer's Responses to Questions

**Comments to the Author**

1. Is the manuscript technically sound, and do the data support the conclusions?

Reviewer #1: Yes

Reviewer #2: Partly

2. Has the statistical analysis been performed appropriately and rigorously? 

Reviewer #1: N/A

Reviewer #2: No

3. Have the authors made all data underlying the findings in their manuscript fully available?

Reviewer #1: Yes

Reviewer #2: Yes

4. Is the manuscript presented in an intelligible fashion and written in standard English?

Reviewer #1: Yes

Reviewer #2: Yes

5. Review Comments to the Author

Reviewer #1: 1. In this study, authors identified the MSX1mutations responsible for the tooth agenesis in two Chinese patients, and explained the reasons why these mutations could lead to tooth agenesis by 3D-structure modeling. On the other hand, the authors reviewed all known MSX1 missense mutations available as that are believed to cause non-syndromic tooth agenesis. So I suggest that the title should be changed to “Two Novel Mutations in MSX1 Causing Oligodontia: Case Report and Literature Review”.

2. A frameshift mutation c.590_594 dup TGTCC was detected in family 2. In order to be more accurate, cloning and sequencing should be used to detect the frameshift mutation in III:1 and II:2 in family 2.

Reviewer #2: The manuscript by Yang et al reported two novel MSX1 mutations responsible for non-syndromic tooth agenesis. The authors also reviewed genotype-phenotype relationship of MSX1 missense mutations via 3D-structural analysis and concluded that MSX1 mutations that altered hydrogen bonding tend to cause a more severe non-syndromic TA phenotype with more missing teeth. The analysis is based on 7 mutations identified in the published literature. Overall the study is interesting and well designed. However, there are some issues that need to be addressed before publication.

1. The author concluded that MSX1 mutations that altered hydrogen bonding tend to cause a more severe non-syndromic TA phenotype with more missing teeth. How did they get this conclusion? Because there was not any statistical genotype-phenotype correlation analysis compared between the MSX1 mutations with or without altered hydrogen bonding. What is the average number of missing teeth in MSX1 mutations with or without altered hydrogen bonding respectively? How many MSX1 mutations without altered hydrogen bonding were included for the statistical analysis and the reason for selection should be well described.

2. The manuscript needs to be carefully revised and well organized. In the Discussion part, from line 211 to line 282, include the descriptions of methods, results and discussion about MSX1 mutations that altered hydrogen bonding seems to cause more severe non-syndromic TA phenotype.

3. All the figures are of poor quality, not clear. They should provide better and higher resolution images to readers.

4. Mutational analysis was only performed in the proband of family 2. The pathogenic mutations of other members in this family are needed to be confirmed, especially his mother (II:2)，since she was also a tooth agenesis patient.

5. How did they get the results of Figure 5 should be well descripted, because that is the key result supporting their conclusion. It should be include in the main body of the manuscript.

6. PLOS authors have the option to publish the peer review history of their article (what does this mean?). If published, this will include your full peer review and any attached files.

Reviewer #1: No

Reviewer #2: No

---

## [Author Response · Author response to Decision Letter 0]

3 Dec 2019

We have revised the parts of the manuscript you proposed to be well organized. The details can be found in "response to reviewers".

---

## [Decision Letter · Decision Letter 1]

17 Dec 2019

Two Novel Mutations in MSX1 Causing Oligodontia

PONE-D-19-23967R1

Dear Dr. Bian,

We are pleased to inform you that your manuscript has been judged scientifically suitable for publication and will be formally accepted for publication once it complies with all outstanding technical requirements.

With kind regards,

Tao Cai, M.D., Ph.D.

Academic Editor

PLOS ONE

Additional Editor Comments (optional):

Reviewers' comments:

Reviewer's Responses to Questions

**Comments to the Author**

1. If the authors have adequately addressed your comments raised in a previous round of review and you feel that this manuscript is now acceptable for publication, you may indicate that here to bypass the “Comments to the Author” section, enter your conflict of interest statement in the “Confidential to Editor” section, and submit your "Accept" recommendation.

Reviewer #2: All comments have been addressed

2. Is the manuscript technically sound, and do the data support the conclusions?

Reviewer #2: Yes

3. Has the statistical analysis been performed appropriately and rigorously? 

Reviewer #2: Yes

4. Have the authors made all data underlying the findings in their manuscript fully available?

Reviewer #2: Yes

5. Is the manuscript presented in an intelligible fashion and written in standard English?

Reviewer #2: Yes

6. Review Comments to the Author

Reviewer #2: The authors have adequately addressed your comments raised in a previous round of review, there is no further comments. The manuscript can be accepted now.

7. PLOS authors have the option to publish the peer review history of their article (what does this mean?). If published, this will include your full peer review and any attached files.

Reviewer #2: No

---

## [Editor Report · Acceptance letter]

27 Dec 2019

PONE-D-19-23967R1 

Two Novel Mutations in MSX1 Causing Oligodontia 

Dear Dr. Bian:

I am pleased to inform you that your manuscript has been deemed suitable for publication in PLOS ONE. Congratulations! Your manuscript is now with our production department. 

With kind regards,

on behalf of

Dr. Tao Cai 

Academic Editor

PLOS ONE